# Determinants and prevalence of modern contraceptive use among sexually active female youth in the Berekum East Municipality, Ghana

**Ebenezer Jones Amoah[1]\*, Thomas Hinneh[2]\*, Rita Aklie[3]**

**1** Ghana Health Service, Municipal Health Directorate, Berekum, Ghana, **2** Johns Hopkins University, School of Nursing, Baltimore, MD, United States of America, **3** Nursing and Midwifery Training College, Pantang, Ghana

\* Hinneh90@gmail.com (TH); ejamoah25@gmail.com (EJA)

## Abstract

### Introduction

Contraceptive use among sexually active women in Ghana remains low despite the efforts by the Ghana Health Service. This development has negative consequences on reproductive health care, particularly among adolescents. This study assessed the prevalence and factors influencing contraceptive use among sexually active young women in the Berekum Municipality, Ghana.

### Method

A community-based cross-sectional analytical study was carried out in Berekum East Municipality among young women between the ages of 15 to 24 years. Using a probabilistic sampling technique, we recruited 277 young women from the four selected communities in the Berekum Municipality based on data available from the Municipal Health Administration. We applied a univariate and multivariate logistic regression analysis to test the associations between the dependent and independent variables at a 95% Confidence interval (CI) and 5% significance (p value = 0.005).

### Results

The modern contraceptive prevalence rate among the study participants was 211 (76%). Contraceptives ever used were emergency contraceptive pills 88 (41.7%) condoms 84 (39.8%), injectables 80 (37.9%) and the rest used the Calendar method 16 (7.58%), withdrawal 15 (7.11%), and implants 11 (5.21%). In the adjusted multivariate logistic regression, Age (AOR = 2.93; 95% CI; 1.29–7.50) p = 0.023, marital status (AOR = 0.08; 95%CI; 0.01–0.91) p = 0.041 and religion (AOR = 0.17; 95% CI; 0.05–0.64) p = 0.009 were significantly associated with contraceptive use. Other determinants such as hearing about contraceptives (AOR = 9.44; 95%CI; 1.95–45.77) p = 0.005, partner opposition (AOR = 33.61; 95%CI; 1.15–985.39) p = 0.041, side effects (AOR = 4.86; 95%CI; 1.83–12.91) p = 0.001, lack of

**Data Availability Statement:** All relevant data are within the paper and its Supporting Information files

**Funding:** The author(s) received no specific funding for this work.

**Competing interests:** The authors have declared that no competing interests exist.

**Abbreviations:** AIDS, Acquire Immune Deficiency Syndrome; AOR, Adjusted Odds Ratio; BMHD, Berekum Municipal Health Directorate; CDC, Center for Disease Control; CHPS, Community–Based Health Planning Services; CHRPE, Committee on Human Research Public and Ethics; CIP, Costed Implementation Plan; CI, Confidence Interval; CPR, Contraceptive Prevalence Rate; ECP, Emergency Contraceptive Pill; GSS, Ghana Statistical Services; HIV, Human Immune Deficiency Virus; KNUST, Kwame Nkrumah University of Science and Technology; IUD, Intrauterine Device; LAM, Lactation Amenorrhea Method; OR, odds ratio; SDGs, Sustainable Development Goals; STI, Sexual Transmitted Infection; UNFPA, United Nations Population Funds; USA, United State of America; WHO, World Health Organization.

knowledge (AOR = 5.41; 95%CI; 1.15–25.42) p = 0.032, and respondents receiving counselling on family planning were significantly associated with contraceptive use (AOR = 4.02; 95% CI;1.29–12.42), p = 0.016.

## Conclusion

Contraceptive use among sexually active women in the Berekum Municipality is higher than the national conceptive prevalence rate. However, factors such as knowledge about the side effects of contraceptive influences contraceptive use among women. Healthcare providers must explore avenues to enhance partner involvement, intensify health education and detailed counselling about contraceptive use to address misconceptions and myths surrounding the side effects of contraceptives.

## Introduction

Unplanned teenage pregnancy is an issue of public health concern since both unintended pregnancy and pregnancy at a young age are linked with negative health consequences for the mother and the infant [1]. About 218 million unwanted pregnancies, fifty-five (55) million unintended births, 138 million abortions and 118 million maternal deaths were averted in developing countries owing to the subscription to family planning methods [2]. Global performance towards achieving the SDG target on family planning and conception stands at 75.7%, with middle and western Africa doing less than 50% [3]. Despite these substantial gains, about 222 women in developing countries still have unmet family planning and contraception needs [4]. Only 21% of women of reproductive age who are married or in cohabitation utilize some type of contraception in Sub-Saharan Africa [5].

In Ghana, unsafe abortion is the second leading cause of pregnancy-related deaths accounting for 20.7% of all-cause put together [6]. Younger women are at a higher risk of dying from abortion-related complications in Ghana [7]. Given that most adolescents indulge in sexual activities even before age 17 years, there is a need to improve adolescent reproductive family planning services to meet the needs of this population [8]. Moreover, younger women have reported more unwanted births than older women in Ghana [9]. The Ghana Statistical Service suggests that about 30% of pregnancies and births occur among young adolescent women in Ghana [10, 11]. Given the increased teenage sexual activity and decreased age of first sex in low-income countries, the use of contraception will be essential in preventing unintended pregnancy and unsafe abortion [12].

The use of contraception in Ghana appears to be low, leading to high rates of unwanted pregnancies, unintended births, unsafe abortions, and maternal mortality [13]. In the past years, Ghana has made strides in eliminating barriers to access to family services through the Costed implementation plan (CIP) initiative [10]. Besides the gains made through the CIP, limited access to family planning services, compounded by limited human resources across health facilities continue to undermine contraceptive utilization in Ghana [10]. Currently, modern contraceptive rates stand at 22.2% among all women in Ghana [14]. Despite the benefit of family planning and contraceptive, uptake of contraceptives among adolescents is affected by numerous socio-cultural and demographic factors [15, 16] which includes cultural beliefs, peer influence, religion, and fear of side effects [17, 18] father's educational background and prior discussion of contraceptive use with a partner [19, 20] age of adolescent, education,

work status, knowledge of ovulatory cycle, visit of health facility, non-youth-friendly health services and marital status [21, 22] spouses or partners made the decision for them to utilize contraception [23]. Other health-related factors reported includes counselling on contraceptive and health provider attitudes [24, 25].

The Berekum East Municipality, between 2018–2019, recorded pregnancy rates of 26.5% and 32% among young people aged 10–24 years [26]. Although studies have reported poor uptake of contraceptives across the various communities in Ghana and mostly these respondents are adolescents, data on family planning utilization among the female youth is limited in the Berekum municipality [11, 27]. As a result, the purpose of this study is to ascertain the prevalence and factors determining contraceptive usage among female youth in Berekum Municipality.

## Materials and methods

### Study design

This cross-sectional analytical study was conducted between June and July 2020 in the Berekum East Municipality.

### Study setting

The Municipality covers a total land area of about 863.3 sq. km. It is bordered to the northeast and North-west by Tain District and Jaman South Districts respectively, South-west by Dormaa East District and Sunyani West District to the southeast. The municipality has the following: two hospitals, a health centre, seven rural clinics, four maternity homes, two private clinics and ten Community Health Planning Services (CHPS) serving a total population of 106,741 people. The Total Fertility Rate for the municipality is 2.8. One facility within the municipality does not provide family planning services. Four sub-municipalities (Berekum central, Zongo, Kato and Mpatasie) and a community within each selected Municipality were chosen for the study.

### Study population

The study population included all female youth aged 15 to 24 years in Berekum Municipality who are sexually active.

Females who lived in the study area for at least six months and consented to the study were included. This enabled us to sample participants who have adequate contextual and cultural understanding of contraceptive use in the locality.

### Sampling techniques

The probability sampling technique was employed. simple random was used to select four sub-municipals out of five sub municipals. the sample size was allocated to each selected sub municipal proportionally based on their expected number of women in reproductive age in each sub municipal. A community was selected randomly from the four sub municipals. The study participant was selected by systematic random sampling method for the households every 8th household. A central reference point, such as a borehole, church, or mosque, was identified in each community. If the chosen household did not have an eligible youth, the next household in the same direction was chosen until the appropriate sample size for the community was reached. In the household, the study was explained to the household members, and approval was sought from the head. Residents between the ages of 15 and 24 were invited individually to a secluded place by the research assistant to inquire about sexual activity, and those

who fulfilled the criteria were asked to participate. After explaining the study's objectives, informed consent was sought, and a questionnaire was administered in English or a local language understandable by both the research assistant and the participant. In the instance of teenagers under the age of 18, consent was sought from a parent/guardian, with assent from the adolescent.

## Data collection techniques

Four research assistants were trained for the data collection from each community. The research assistant visited the chosen communities to select households and recruit participants. A structured questionnaire was used to collect data. The first section of the questionnaire collected information of participants' socio-demographic characteristics including age, marital status, educational level, religion, ethnicity, place of residence, and person living with. The second section of the questionnaire focused on sexual behaviors and knowledge of modern contraceptives. The third section elicited information about potential barriers and facilitators of modern contraceptive use. Modern contraceptive methods include contraceptive pills, implants, injectables, intrauterine devices (IDU), female and male condoms, female and male sterilization, vaginal barrier methods (including the diaphragm, cervical cap, and spermicidal agents), lactational amenorrhea method (LAM), emergency contraception pills. Knowledge was measured as ever heard (Yes/No), use of modern contraceptives (Yes/ No) and mention of at least one of the modern contraceptive methods.

## Sample size calculation

The study population included all youth aged 15 to 24 years in Berekum Municipality. The sample size was calculated in Epi Info, version 7.1.1.14 (Centers for Disease Control and Prevention, Atlanta, GA, USA). To achieve 80% power, we allowed 95% confidence intervals (CIs) and a 5% margin of error and accounted for 10% contingency. We calculated that a sample size of 277 would have adequate power (80%) to detect factors with use of Epi info StatCalc.

## Statistical analysis

Descriptive statistics were adopted to describe the factors associated with contraceptive use using the statistical software STATA version 15. A Chi-square test was used to measure the association or relationship between the outcome variable (contraceptive use) and the explanatory variables. Regression analysis (logistic regression) was employed to assess the odds ratio (ORs) of the factors associated with contraceptive use at a 95% Confidence interval (CI) and 5% significance (p value = 0.005).

## Ethical consideration

Ethical approval for the study was obtained from the Committee on Human Research, Publications and Ethics (CHRPE) of KNUST with reference number CHRPE/AP/306/20. Written consent was sought from the study participants after an informed consent form was read and explained to them. While those below 18 years, consent from parents/guardian was sorted before obtaining assent from each respondent. We assured privacy and confidentially of information collected during the process.

## Results

### Socio-demographic characteristics of the respondents

A total of 277 females were approached to participate in the current study. The mean age of the respondents was (mean SD) 19 ± 2.6 years. More than half,148 (53.4%) of the respondents were between the ages of 15–19 years. Almost half of the respondents 114 (41.2%) and 107 (38.6%) have had Primary/JHS and Secondary education respectively. The majority 170 (61.4%) of the study population were students and a vast majority 202 (73%) were from the Akan tribe. About 223 (81%) and 234 (84.48%) of the respondents were single and were Christians respectively. Most respondents 169 (61.0%) reside in rural areas and 125 (45.1%) live with both parents (Table 1).

### Knowledge of contraceptive uses

Almost all 253 (91%) of the respondents ever heard of contraceptives or family planning. Respondents indicated multiple reasons for contraceptive use which include delaying

**Table 1. Socio-demographic characteristics of participants.**

| Variables | Freq. | Percent |
|---|---|---|
| Mean Age (SD) | | 19 (2.6) |
| **Age group** | | |
| 15–19 | 148 | 53.43 |
| 20–24 | 129 | 46.57 |
| **Education level of Respondent** | | |
| No formal Education | 30 | 10.83 |
| Primary / JHS | 114 | 41.16 |
| Secondary | 107 | 38.63 |
| Tertiary | 26 | 9.39 |
| **Occupation of Respondent** | | |
| Apprentice | 55 | 19.86 |
| Employed | 20 | 7.22 |
| Students | 170 | 61.37 |
| Unemployed | 32 | 11.55 |
| **Ethnicity** | | |
| Akan | 202 | 72.92 |
| Northern ethnics | 75 | 27.08 |
| **Marital status** | | |
| Married/ co-habiting | 54 | 19.49 |
| Single | 223 | 80.51 |
| **Religion** | | |
| Christianity | 234 | 84.48 |
| Islamic | 43 | 15.52 |
| **Residence status** | | |
| Both Parent | 125 | 45.13 |
| Guardian | 27 | 9.75 |
| Live alone | 25 | 9.03 |
| Partner | 21 | 7.58 |
| Single Parent | 79 | 28.52 |
| **Place of Residence** | | |
| Rural | 169 | 61.01 |
| urban | 108 | 38.99 |

**Table 2. Knowledge of respondent on contraceptive.**

| Variables | Freq. | Percent |
|---|---|---|
| **Ever heard about FP** | | |
| No | 24 | 8.66 |
| Yes | 253 | 91.34 |
| **Uses of FP methods** * | | |
| To prevent pregnancy | 41 | 16.47 |
| To delay pregnancy | 219 | 87.95 |
| To space up birth | 58 | 23.29 |
| To prevent STI/HIV | 51 | 20.48 |
| **FP methods heard about** * | | |
| Male condom | 131 | 51.98 |
| Female condom | 33 | 13.1 |
| Injectables | 129 | 51.19 |
| Implants | 32 | 12.7 |
| Sterilization | 8 | 3.17 |
| Emergency Contraceptive Pill | 82 | 32.54 |
| IUD | 10 | 3.97 |
| COC and POP pills | 66 | 26.19 |
| Calendar Method | 28 | 11.11 |
| Lactational Amenorrhea Method | 1 | 0.4 |
| **Source Information** * | | |
| Radio/Television | 86 | 34.13 |
| Teacher | 44 | 17.46 |
| Friend | 117 | 46.43 |
| Health worker | 144 | 57.14 |
| Partner | 37 | 14.68 |
| Parent | 11 | 4.37 |
| Other relatives | 7 | 2.78 |
| **Who is eligible to use FP** | | |
| Adult only | 43 | 15.52 |
| All sexually active person | 206 | 74.37 |
| Married couples | 28 | 10.11 |
| **Women who use FP are promiscuous** | | |
| Don't know | 79 | 28.52 |
| No | 165 | 59.57 |
| Yes | 33 | 11.91 |

* Multiple responses

pregnancy 219 (88%), preventing STIs and HIV/AIDS 51 (20%), spacing up birth 58 (23%) and 41 (16%) stated that contraceptives are used to prevent pregnancy (Table 2).

Among the respondents the most known contraceptives were male condom 131(52%), Injectables129 (51.2%), Emergency contraceptive Pill (ECP) 82 (32.5%), Pills (Microgynon and Microlut) 66 (26.2%) and Implants 32 (12.7%). Other known family planning methods includes Natural/ calendar method, IUD, Male sterilization, and LAM. A vast majority 206 (74%) of the respondent indicated that all sexually active persons should use contraceptives. About 60% (165) of the respondent believe women who use contraceptives are not

promiscuous. Among all respondents, majority received source information from a health worker 144 (57.14%), friends 117 (46.43%) and Radio/Television 86 (34.13%) (Table 2).

## Contraceptive utilization among sexually active female youth

Prevalence of contraceptive use among the study participants was significantly high as a little over 76% (211) of the respondents reported ever using any modern contraceptive. However, a little over 63% (132) used contraceptive methods in their first sexual encounter (Table 3). The common methods of contraceptives used before were emergency pills 88 (41.2%), condoms 84 (39.8%) injectables 80 (37.9%) implants 11 (5.21%) and IUDs 1 (0.47%), also traditional methods such as calendar 16 (7.6%) and withdrawal methods 15 (7.1%) used before (Fig 1). At the time of the study, about 151 (72%) of the respondents were currently using some form of modern contraceptives which were Injecta ble 56 (37.1%), ECP 47 (31.13%) and Male Condoms 9 (19.2%). However, only 10 (6.6%) are LARC (Implants) users. Half 75 (50%) of the respondents get their source of contraceptives from a Health Facility, whiles 48 (32%) get theirs from a Drug store/pharmacy shop. Most 89 (42%) of the respondents have been using contraceptives for about 1–11 months and 95 (45%) used these contraceptives occasionally during sexual intercourse. About 33% (90) of the respondents had ever been pregnant and more than half 56% (50) of those pregnancies resulted in livebirth whiles 29 (32%) resulted in induced abortion. More than half 48 (53%) of the respondents had their first pregnancy between the ages of 11 and 18 years. The number of sexual partners ever had included 1 (47.6%), 2 (25.6%), 3 (18.4%) and 23 (8.3%) ever had four sex partners. It was evident that majority of respondents had their first sex between the ages of 10–16 years (Table 3).

## Socio-cultural and health-related factors influencing contraceptive use among sexually active

The study identified several socio-cultural and health-related factors that are associated with contraceptive use. Among these factors, the most significant predictors were minimal side effects (54.18%), receipt of counselling on contraceptives (42.55%), partner support (36.73%), religious beliefs (36.36%), lack of knowledge (21.45%), attitude of service providers (26.18%), and parental support (11.64%) etc. (Fig 2).

## Determinants of modern contraceptive use among sexually active young women

As presented in Table 4, In a univariate analysis, the following independent variables were significantly associated; Age group (p<0.001), marital status (p = 0.002), religion (p = 0.027), age at first sex(p = 0.001), number of sex partners(p = 0.003), feeling pressure to have sex (p = 0.033), ever heard about FP(p<0.001), partners opposition (p = 0.039), side effect (p = 0.029), lack of knowledge (p = 0.038), partners support (p = 0.036), parental support (p = 0.007), counselling received on contraceptive (p = 0.045) were all significantly associated with contraceptive use.

Adjusting for the confounders of the variables on the dependent variable in the model, the multivariate logistic regression revealed a statistically significant association between age group, marital status, religious affiliation, heard about contraceptives, partner opposition, side effect, lack of knowledge and counselling received. It was evident that Age was a factor in contraceptive use. older youth (20–24) were 2.93 times more likely to use contraceptives than those at the adolescent stage (15–19) (AOR = 2.93; 95% CI; 1.29–7.50), p = 0.023. Marital status was significantly associated with contraceptive use. Respondents who were single were less

**Table 3. Contraceptive utilization among respondents.**

| Variables | Freq. | Percent |
|---|---:|---:|
| **Ever use FP before** | | |
| No | 66 | 23.83 |
| Yes | 211 | 76.17 |
| **Currently using FP methods?** | | |
| No | 60 | 28.44 |
| Yes | 151 | 71.56 |
| **Which method currently using** | | |
| Emergency Contraceptive Pills | 47 | 31.13 |
| Implants | 10 | 6.62 |
| Injectables | 56 | 37.09 |
| Male Condom | 29 | 19.21 |
| Withdrawal | 9 | 5.96 |
| **Source of FP** | | |
| Drug store/ Pharmacy shop | 48 | 32.21 |
| Friend | 3 | 2.01 |
| Health facility | 75 | 50.34 |
| Not Applicable | 8 | 5.37 |
| Partner | 15 | 10.07 |
| **How often do you use FP** | | |
| Every time | 81 | 38.39 |
| Just once | 35 | 16.59 |
| once a while | 95 | 45.02 |
| **How long have you use FP** | | |
| 1 to 11 months | 89 | 42.18 |
| 1 to 2 years | 69 | 32.70 |
| 3 years above | 20 | 9.48 |
| Less than 1 month | 33 | 15.64 |
| **FP method used during first sex** | | |
| No | 79 | 37.44 |
| Yes | 132 | 62.56 |
| **Age at first sex** | | |
| 10–16 | 151 | 54.51 |
| 17–23 | 126 | 45.49 |
| **Number of sex partners** | | |
| 1 | 132 | 47.65 |
| 2 | 71 | 25.63 |
| 3 | 51 | 18.41 |
| 4 | 23 | 8.30 |
| **Ever been pregnant before** | | |
| No | 187 | 67.51 |
| Yes | 90 | 32.49 |
| **Pregnant for how many times** | | |
| 1 | 67 | 74.44 |
| 2 | 18 | 20.00 |
| 3 | 5 | 5.56 |
| **Age at first pregnancy** | | |
| 11–18 | 48 | 53.33 |

(*Continued*)

**Table 3.** (Continued)

| Variables | Freq. | Percent |
|---|---|---|
| 19–24 | 42 | 46.67 |
| **First pregnant outcome** | | |
| Currently pregnant | 3 | 3.33 |
| Induced abortion | 29 | 32.22 |
| Miscarriage | 8 | 8.89 |
| Resulted in livebirth | 50 | 55.56 |
| **Feel pressure to have sex** | | |
| No | 207 | 74.73 |
| Yes | 70 | 25.27 |

likely to use contraceptives as compared to those who were married (AOR = 0.08; 95%CI; 0.01–0.91), p = 0.041 (Table 4).

Respondents affiliated with the Islamic religion were less likely to use contraceptives as compared to those affiliated with Christianity (AOR = 0.17; 95%CI; 0.05–0.64), p = 0.009.

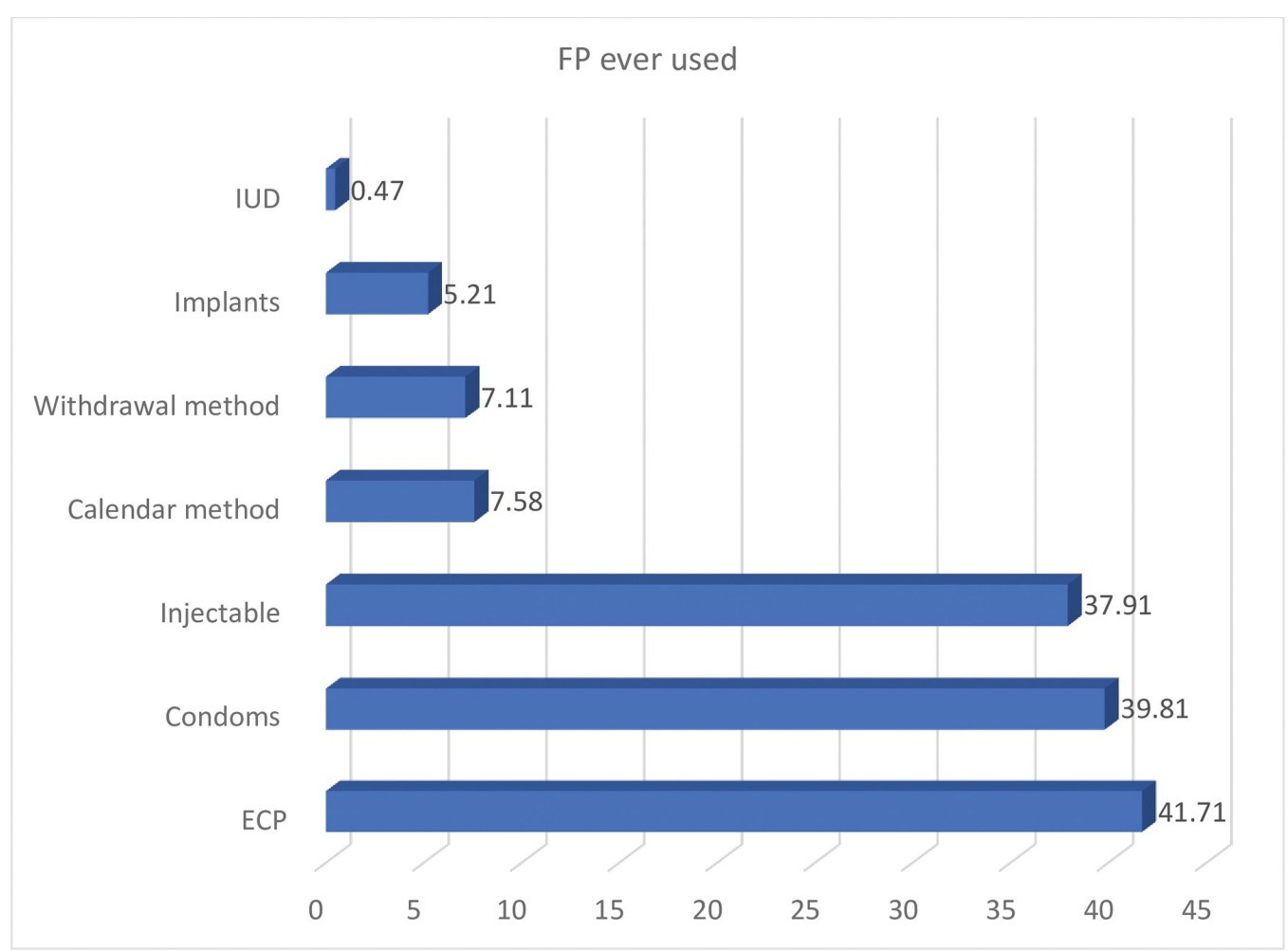

**Fig 1. FP method ever used before.**

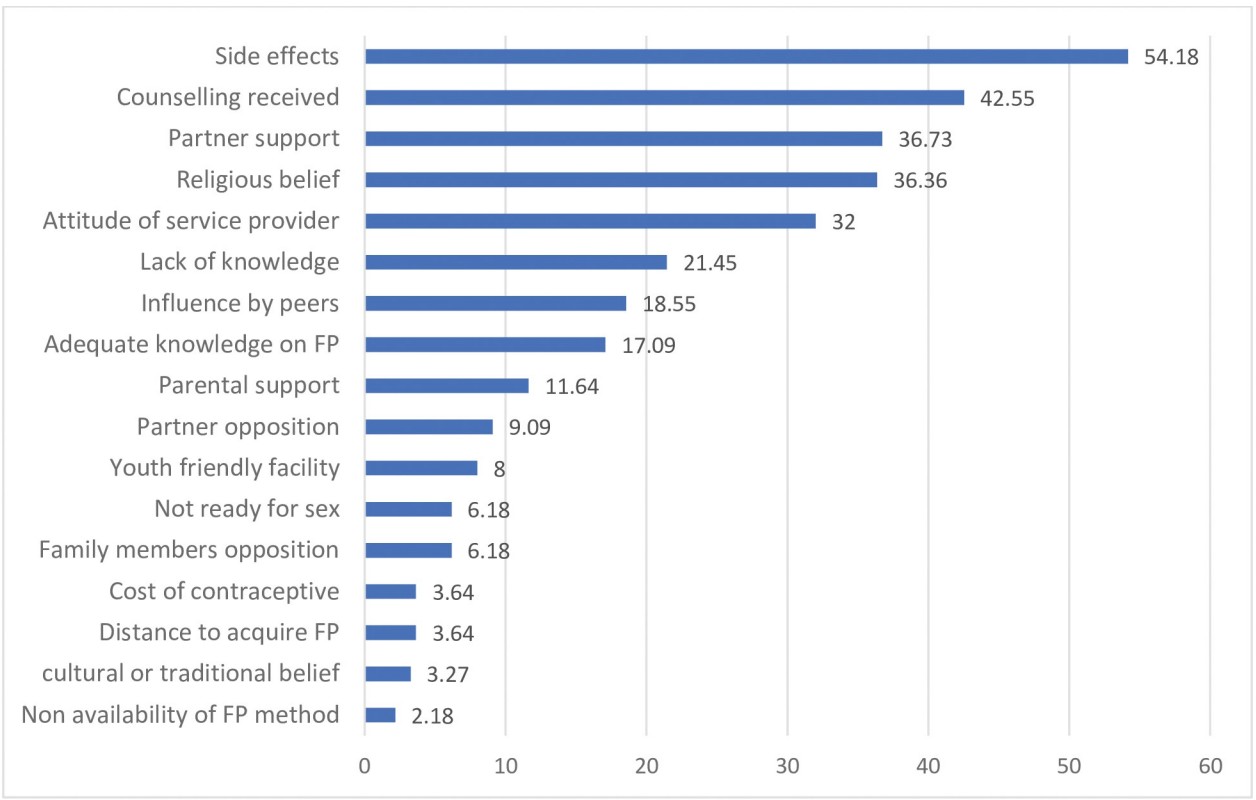

**Fig 2. Socio-cultural and health-related factors influencing contraceptive use.**

Respondents who have heard about contraceptives before were 9.44 more likely to use contraceptives as compared to respondents who have never heard about contraceptives (AOR = 9.44; 95%CI; 1.95–45.77), p = 0.005. Partner opposition as a factor was 33.61 more likely to influence contraceptives as compared to those who did not (AOR = 33.61; 95%CI; 1.15–985.39), p = 0.041. Respondents reporting side effects were significantly associated with 4.86 times likely to use contraceptives (AOR = 4.86; 95%CI; 1.83–12.91), p = 0.001. Respondents with a lack of knowledge of contraceptives were 5.41 times more likely to use contraceptives compared to those respondents with knowledge of contraceptives (AOR = 5.41; 95%CI; 1.15–25.42), p = 0.032. Respondents receiving counselling on family planning were 4.02 more likely to use contraceptives compared to those who do not receive counselling on family planning (AOR = 4.02; 95% CI;1.29–12.42), p = 0.016 (Table 4).

## Discussion

The benefit of contraceptive use is critical to safe adolescent and sexual reproductive health. Credible information and unlimited access to quality and culturally appropriate adolescent services influence the uptake of modern contraceptives. The study revealed that most of the participants reported using modern contraceptive methods, and this prevalence was higher than that reported in previous studies of sexually active young women in Ghana [11, 28]. However, a similar study reported low contraceptive use among sexually active unmarried adolescent girls (10–19 years; 35.6%) and young women (15–24 years; 49.0%) [29]. The widespread availability and affordability of certain modern contraceptive methods, such as ECPs and condoms sold in various locations like chemical seller's shops, restaurants, and supermarkets,

**Table 4. Univariate and multivariate regression model on determinants of modern contraceptive use among sexually active female youth.**

| Variables | Crude Odds Ratio | | Adjusted Odds Ratio | |
| --- | --- | --- | --- | --- |
| | COR (95% CI) | p-value | AOR (95% CI) | p-value |
| **Age group** | | | | |
| 15–19 | 1 | | 1 | |
| 20–24 | 2.96 (1.62–5.42) | **<0.001** | 2.93 (1.20–7.50) | **0.023** |
| **Marital status** | | | | |
| Married/ cohabiting | 1 | | 1 | |
| Single | 0.15 (0.05–0.50) | **0.002** | 0.08(0.01–0.91) | **0.041** |
| **Religion** | | | | |
| Christianity | 1 | | | |
| Islamic | 0.46 (0.23–0.917) | **0.027** | 0.17 (0.05–0.64) | **0.009** |
| **Age at first sex** | | | | |
| 10–16 | 1 | | 1 | |
| 17–23 | 2.80 (1.53–5.12) | **0.001** | 1.35 (0.40–4.55) | 0.627 |
| **Number of sex partners** | | | | |
| 1 | 1 | | 1 | |
| 2 | 3.33 (1.51–7.32) | **0.003** | 1.17 (0.33–4.13) | 0.808 |
| 3 | 1.76 (0.82–3.76) | 0.146 | 0.23 (0.05–1.01) | 0.052 |
| 4 | 3.22 (0.91–11.43) | 0.07 | 0.74 (0.07–7.39) | 0.798 |
| **Feel pressure to have Sex** | | | | |
| No | 1 | | 1 | |
| Yes | 2.23 (1.10–4.65) | **0.033** | 3.19(0.96–10.62) | 0.058 |
| **Ever heard about FP** | | | | |
| No | 1 | | 1 | |
| Yes | 22.50 (7.34–68.96) | **<0.001** | 9.44 (1.95–45.77) | **0.005** |
| **Who is eligible to use FP** | | | | |
| Adult only | 1 | | 1 | |
| All sexually active person | 2.90 (1.45–5.81) | **0.003** | 0.91(0.25–3.25) | 0.884 |
| Married couples | 2.16 (.76–6.16) | 0.15 | 1.31(0.26–6.61) | 0.744 |
| **Partner opposition** | | | | |
| No | 1 | | 1 | |
| Yes | 8.43 (1.12–63.58) | **0.039** | 33.61(1.15–985.39) | **0.041** |
| **Side effects** | | | | |
| No | 1 | | 1 | |
| Yes | 1.87 (1.07–3.27) | **0.029** | 4.86(1.83–12.91) | **0.001** |
| **Lack knowledge** | | | | |
| No | 1 | | 1 | |

(*Continued*)

**Table 4.** (Continued)

| | Crude Odds Ratio | | Adjusted Odds Ratio | |
|---|---|---|---|---|
| **Variables** | **COR (95% CI)** | **p-value** | **AOR (95% CI)** | **p-value** |
| Yes | 2.34 (1.05–5.23) | **0.038** | 5.41(1.15–25.42) | **0.032** |
| **Attitude of Health Provider** | | | | |
| No | 1 | | 1 | |
| Yes | 3.81 (1.65–8.78) | **0.002** | 2.15(0.56–8.27) | 0.264 |
| **Partner support** | | | | |
| No | 1 | | 1 | |
| Yes | 1.94 (1.05–3.59) | **0.036** | 1.06(0.37–3.01) | 0.909 |
| **Parental support** | | | | |
| No | 1 | | 1 | |
| Yes | 0.35(0.16–0.75) | **0.007** | 0.31(0.07–1.42) | 0.132 |
| **Counselling received** | | | | |
| No | 1 | | 1 | |
| Yes | 1.82 (1.01–3.27) | **0.045** | 4.02(1.29–12.42) | **0.016** |

have contributed to the high prevalence of modern contraceptive use among female youth. These results imply a positive trend in the use of contraception among young women in Ghana.

The emergency contraceptive pill, condoms, and injectables were identified as the most commonly preferred modern contraceptive methods among the participants, which is consistent with previous research highlighting these options as popular choices among unmarried adolescents and young women [25, 29, 30]. The widespread use of these methods may be due to their greater accessibility, availability, perceived efficacy, and convenience of use, which come with fewer side effects than injectables and implants.

Several studies have observed comparable patterns in Ghanaian individuals' understanding and awareness of modern contraceptive methods [23, 31–33]. In the current study, evidence suggests that there is a positive correlation between individuals' awareness and knowledge of modern contraceptive methods and their adoption of these methods. This is supported by a study conducted among female undergraduates in Tanzania, which found that participants displayed a high level of awareness and knowledge regarding modern contraceptive methods [34]. Improved access to information on reproductive health, as well as the use of behavior change communication and social marketing techniques, may have contributed to the observed developments. These factors have helped to increase awareness and knowledge of modern contraceptive methods, leading to a greater uptake of these methods among individuals [35, 36].

The study found that respondents reported using modern contraceptives for various reasons, including delaying pregnancy, spacing out births, preventing sexually transmitted infections (STIs) and HIV/AIDS, and preventing unintended pregnancy. This finding is consistent with previous research, such as the Katama and Hibstu study (2016) in South Ethiopia, where many participants reported using modern contraceptives to avoid unintended pregnancy, space out or limit conception, and prevent STIs and HIV/AIDS [37, 38]. This suggests that these reasons for modern contraceptive use are prevalent across different geographic locations

and populations, underscoring the universal importance of accessible and comprehensive reproductive health services.

One possible reason for the high percentage of delaying pregnancy is that individuals may want to establish their careers, achieve financial stability, or complete their education before starting a family. Furthermore, many individuals may want to ensure that they are emotionally and mentally ready for parenthood before deciding to conceive.

Our study found a statistically significant association between age group and modern contraceptive use. Older youth were more likely to use modern contraceptives compared to those in the adolescent stage. This finding is consistent with a similar study, which found that girls aged 15–19 were more likely to practice contraception compared to those aged 10–14 [39]. Other studies from different parts of Africa also confirm this finding [40, 41]. The higher likelihood of contraceptive use among older youth in Africa may be attributed to differences in social norms around sexual behavior and contraceptive use, as well as greater access to reproductive health services compared to younger adolescents. Older youth may have more autonomy and agency in making decisions about their reproductive health and have greater opportunities to obtain modern contraceptives.

Single women in our study were much less likely to use contraceptives compared to those who were married or cohabiting. This finding is consistent with previous research, which has shown that marital status is a significant factor in contraceptive use among women, especially teenagers [21, 39, 42]. Married women may have a greater desire to control their fertility and plan their family size due to various reasons such as financial stability, career aspirations, and a desire to provide the best possible care for their children. They may also have access to reproductive health services, including information and counselling on contraceptive methods, and may have more social support for contraceptive use from their partners and families. Our study found a significant difference in contraceptive use between respondents affiliated with the Islamic religion and Christianity. Islamic respondents reported lower use of contraceptives compared to their Christian counterparts, which is consistent with previous studies conducted in Ghana [17, 43]. Kumbeni et al. (2019) presented a divergent perspective on the topic of contraceptive use by revealing significant variations in usage rates between Muslims and Christians. Specifically, their study found that Muslims reported greater rates of contraceptive use than Christians, contrasting with prior research [39]. Religious groups that provide sex education on modern contraceptives tend to have higher contraceptive use than those that do not [44]. Also, the differences could be religious interpretations and beliefs regarding family planning and contraception. Some Christians may view contraception as a means of responsible parenthood, while others may believe that it goes against God's plan for procreation.

The reported side effects of contraceptives have been identified as a common obstacle to contraceptive use, as supported by various studies. Concerns about potential side effects can have a notable influence on an individual's choice to utilize contraceptives [17, 22, 25, 39]. These findings underscored the importance of good education and counselling in reducing lack of contraceptive desire among sexually active young women. Recognizing and understanding these challenges is critical for developing effective solutions to address the poor contraceptive usage among youth in our societies and around the country.

In the study, partners who opposed contraceptive use were found to have a significant influence on whether or not the respondent used contraceptives, which is consistent with previous studies [39, 45]. On the contrary, a study conducted by Asiedu et al. (2020) [23], indicated that husbands or partners agreeing on the use of modern contraceptives influenced the usage of modern contraceptives among women. This was the case in another study conducted in Ghana where contraceptive use was higher among women if their spouses or partners support it [46].

This development may trace its roots to typical African societal norms, where a man is the head of the home and decides all matters, including their spouses' reproductive health issues [47]. In some cultures, having many children is a sign of wealth and prosperity and contraceptive use is against religious or moral principles.

In line with previous research, our study has demonstrated that a lack of knowledge and counseling regarding contraceptives, as well as a lack of credible and unbiased sources of information, can significantly hinder their utilization. The evidence highlights the importance of comprehensive and impartial contraceptive education and counseling to increase the likelihood of contraceptive use [48–50]. Additionally, a study conducted in the US found that women who got contraceptive counselling were nearly seven times more likely to use highly effective contraception than women who did not [51]. A lack of knowledge, counselling, and access to credible sources of information on contraception can negatively impact contraceptive use. Individuals may not fully understand the benefits and potential risks associated with different methods of contraception, may not receive guidance from healthcare providers, and may rely on unreliable sources of information. Ultimately, this can increase the risk of unintended pregnancy or sexually transmitted infections.

Inadequate knowledge of contraceptives is linked to wrong beliefs about the risk and negative effects of using them, improper or irregular usage, as well as method discontinuation [52]. This implies that healthcare providers must give evidence-based information on contraception methods while also listening to women's views and thoughts so that they may make well-informed choices regarding the most appropriate contraceptive methods for them. Furthermore, the fact that worries about side effects are becoming widespread shows that contraceptive products are becoming more widely available and that counselling about contraceptive alternatives is an important component of good service delivery.

## Strengths and limitations of the study

In Ghana, most women prefer to conserve privacy with the use of contraceptives. Also, the estimation of contraceptive use was based on participants self -reports. These factors could potentially cause social desirability biases and affect the actual prevalence of contraceptive use in Berekum Municipality. Despite these limitations, this study provides insights on contraceptives in the Berekum municipality which may be useful in improving reproductive health among young women in Ghana.

## Conclusion

The contraceptive prevalence rate among sexually active youth in Berekum municipality was high

Emergency contraceptives, condoms and injectables were the most preferred option compared to other methods. The study also highlighted potential determinants of contraceptives among the female youth in the Berekum Municipality.

There is need for stakeholders, and healthcare providers to focus on addressing factors influences contraceptive use among sexually active women. The Ghana Health Service must initiate measures to fully rectify adolescent health as part of the mainstream school health activities in Senior high Schools in Ghana as a strategy to improving sexual health education Ghana.

## Supporting information

**S1 Dataset.**
(XLSX)

## Acknowledgments

The authors are grateful to the Bono Regional Health Directorate for granting us permission to conduct the study. We are also thankful for the Berekum municipal health directorate, especially the staff who were recruited as research assistants for their cooperation and support.

## Author Contributions

**Conceptualization:** Ebenezer Jones Amoah.

**Data curation:** Ebenezer Jones Amoah, Rita Aklie.

**Formal analysis:** Thomas Hinneh.

**Methodology:** Ebenezer Jones Amoah, Thomas Hinneh, Rita Aklie.

**Resources:** Ebenezer Jones Amoah, Thomas Hinneh, Rita Aklie.

**Validation:** Ebenezer Jones Amoah, Thomas Hinneh.

**Visualization:** Ebenezer Jones Amoah, Thomas Hinneh, Rita Aklie.

**Writing – original draft:** Ebenezer Jones Amoah, Thomas Hinneh, Rita Aklie.

**Writing – review & editing:** Ebenezer Jones Amoah, Thomas Hinneh, Rita Aklie.

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
