## [Decision Letter · Decision Letter 0]

10 Apr 2023

PONE-D-23-04760Determinants and prevalence of modern contraceptive use among sexually active women in the Berekum East Municipality, Ghana.PLOS ONE

Dear Dr. Amoah,

Thank you for submitting your manuscript to PLOS ONE. After careful consideration, we feel that it has merit but does not fully meet PLOS ONE’s publication criteria as it currently stands. Therefore, we invite you to submit a revised version of the manuscript that addresses the points raised during the review process.

We look forward to receiving your revised manuscript.

Kind regards,

Akaninyene Eseme Bernard Ubom, MBBS, MWACS, OMI Fellow

Academic Editor

PLOS ONE

Journal Requirements:

2. In the ethics statement in the Methods, you have specified that verbal consent was obtained. Please provide additional details regarding how this consent was documented and witnessed, and state whether this was approved by the IRB

Additional Editor Comments:

Abstract

Results:

-Absolute numbers should be reported in addition to percentages.

-Where significant association is reported, p values should be quoted.

Manuscript Text:

Materials and Methods (sampling techniques): Page 6, line 125: "The probability sampling technique was employed" Multistage sampling is mentioned in the Abstract. Please clarify/reconcile.

Results

-Where percentages are reported, absolute numbers should also be reported.

-Page 11, Lines 210-211: "The common methods of contraceptives.....,also traditional methods such as calendar and withdrawal methods used before." What were the absolute numbers and percentages of respondents that used these traditional methods?

-Page 12, Lines 221-222: "Shockley majority of respondents had their first sex between the ages of 10-16 years (Table 3)." Please replace "shockingly", this is academic writing. Indicate the absolute number and percentage of women

-Table 2: What is the difference between "to prevent pregnancy", "to delay pregnancy" and "to space birth"? What do you mean by "multiple response"? indicate as a footnote under the table.

-Table 3: Correct "Your source FP supply" to "Source of FP" What does "Not applicable" mean? "Just once" response under "how often do you use FP" does it mean the respondents have used FP only once in their lifetime? Please clarify

-Pages 13-14, lines 231-234: "Socio-cultural and health-related predictors of contraceptive use were minimal side effects (54.18%), counselling received on contraceptives (42.55%), partners support (36.73%), religious belief (36.36%), lack of knowledge (21.45%), the attitude of a service provider (26.18%), parental support (11.64%) etc (Fig 2)." This is not clear, please recast. What exactly is figure 2 about, please explain.

-Page 14, lines 239-247: "As presented in Table 4, In a univariate analysis, the following independent variables were significantly associated;......lack of knowledge and counselling received." Please quote p values for every variable reported as significantly associated with contraceptive use.

-Page 14, lines 247-248: "It was evident that Age was a factor in contraceptive use. older youth were 2.93 times more likely to use contraceptives than those at the adolescent stage." Please clarify which age groups were classified as "older youth" and "adolescent"

-Page 14, lines 250-253: "Respondents who were single were 92 times less likely to use contraceptives as compared to those who were married (AOR=0.08; 95%CI; 0.01-0.91), p=0.041 (Table 4). Respondents affiliated with the Islamic religion were 83 times less likely to use contraceptives as compared to those affiliated with Christianity (AOR=0.17; 95%CI; 0.05-0.64), p=0.003" Please reconcile the figures 92 and 83 quoted with the AOR.

-Table 4: How was "pressure to have sex" assessed?

Discussion:

-Page 16, lines 272-274: "However, similar studies reported low contraceptive use among sexually active unmarried adolescent girls (15–19 years; 35.6%) and young women (20–24 years; 49.0%) [29]" Similar studies is mentioned by only one is cited/referenced. The adolescent age group is 10-19 and youth "15-24"; please reconcile with 15-19 and 20-24 years quoted here.'

-The Discussion should be rewritten. Results should not be re-reported verbatim in the Discussion but discussed. Beyond simply comparing the results of this study with those of other studies, authors should explain possible reasons for their findings with relevant literature references. Results not initially reported in the results section should not be introduced for the first time in the Discussion

Reviewers' comments:

Reviewer's Responses to Questions

**Comments to the Author**

1. Is the manuscript technically sound, and do the data support the conclusions?

Reviewer #1: Partly

2. Has the statistical analysis been performed appropriately and rigorously? 

Reviewer #1: Yes

3. Have the authors made all data underlying the findings in their manuscript fully available?

Reviewer #1: Yes

4. Is the manuscript presented in an intelligible fashion and written in standard English?

Reviewer #1: Yes

5. Review Comments to the Author

Reviewer #1: 1. The title of the study needs to be modified to reflect the actual study population. The study was titled ‘Determinants and prevalence of modern contraceptive use among sexually active women’. This gives an impression that all sexually active women in reproductive age group should be part of the study. However, the Methodology and study in itself was limited to youths (people aged 15-24 years by United Nation’s definition).

2. How does the findings from this study speak to a GLOBAL AUDIENCE with respect to contraceptive prevalence among young people?

3. The verbal interpretation of the odds ratio for Married respondents and Islamic faithfuls does not align with the data on the regression table. It was reported that Married youths and Islamic faithfuls were 92 and 83 times less likely to use contraceptives (respectively). This statement can’t be deduced from the table. Kindly reconcile

4. The discussion was quite lacking in intellectual content. It appeared to be more of a repetition of the results. I will be good to revisit the discussion

5. The references were not properly written. They should be rewritten according to Vancouver guidelines.

6. PLOS authors have the option to publish the peer review history of their article (what does this mean?). If published, this will include your full peer review and any attached files.

Reviewer #1: No

---

## [Author Response · Author response to Decision Letter 0]

3 May 2023

Journal Submissions Rebuttal Letter

Akaninyene Eseme Bernard Ubom, MBBS, MWACS, OMI Fellow

Academic Editor

PLOS ONE

4th May 2023.

Dear Dr Akaninyene,

Re: Resubmission of a manuscript (PONE-D-23-04760)

Thank you for inviting us to submit a revised draft of our manuscript entitled, "Determinants and prevalence of modern contraceptive use among sexually active female youth in the Berekum East Municipality, Ghana” to PLOS ONE for publication. We also appreciate the time and effort you and other reviewers have dedicated to providing insightful feedback on ways to strengthen our paper. Thus, it is with great pleasure that we resubmit our manuscript for further consideration. We have incorporated changes that reflect the detailed suggestions you have graciously provided. We also hope that our edits and responses below satisfactorily address all the issues and concerns you have noted.

Below we provide the point-by-point responses. All modifications in the manuscript have been highlighted in red and are in track changes.

Yours Sincerely,

Ebenezer Jones Amoah (corresponding author)

ejamoah25@gmail.com

Journal Requirements

Comment: In the ethics statement in the Methods, you specified that verbal consent was obtained. Please provide additional details regarding how this consent was documented and witnessed, and state whether this was approved by the IRB

Response: Thank you for the observation, we initially explained the intent and procedures of data collection of the research to the participant verbally. Upon agreement to part-take, we administered the consent form. Given the fact that some of the participants could not read and write, they were asked to thumbprint. The consent form was sent in addition to the proposal and approval was given by the Kwame Nkrumah University of Science and Technology IRB to start data collection. Sorry for the omission. 

Additional Editor Comments:

Abstract

Results:

Comment: Absolute numbers should be reported in addition to percentages.

Response: Thank you for these observations. We have added the absolute numbers in addition to the percentages in the abstract in lines 47 – 57.

Comment: Where significant association is reported, p values should be quoted.

Response: Thank you once again for these observations. p-values have been quoted to show how significantly variables are associated in the abstract in lines 51 - 57

Manuscript Text:

Comment: Manuscript Text: Materials and Methods (sampling techniques): Page 6, line 125: "The probability sampling technique was employed" Multistage sampling is mentioned in the Abstract. Please clarify/reconcile

Response: Thank you for the observation, Page 2, line 39 has been reconciled from multistage to probabilistic to align with page 6, line 127.

Results

Comment: 

Where percentages are reported; absolute numbers should also be reported. 

Response: Thank you for the observation. Where percentages were reported in the result session, absolute numbers have also been reported.

Comment: Page 11, Lines 210-211: "The common methods of contraceptives.....,also traditional methods such as calendar and withdrawal methods used before." What were the absolute numbers and percentages of respondents that used these traditional methods? 

Response: Thank you for the observation. On page 11, line 214 The percentages and absolute numbers of calendar-16 (7.6%), and withdrawal-15 (7.1%) have been reported. Thank you for the observation.

Comment: Page 12, Lines 221-222: "Shockley majority of respondents had their first sex between the ages of 10-16 years (Table 3)." Please replace "shockingly", this is academic writing. Indicate the absolute number and percentage of women.

Response: Thank you very much for alerting us. Sorry for using such a word as “shockingly” in line 225. “shockingly” has been replaced with “It was evident that the.” Also, the absolute number and percentage have been indicated.

Comment: Table 2: What is the difference between "to prevent pregnancy", "to delay pregnancy" and "to space birth"?

Response: Thank you for the enquiry. Please, while all three phrases relate to family planning, they have different meanings and objectives. "To prevent pregnancy" is about avoiding pregnancy altogether, "to delay pregnancy" is about postponing pregnancy until a later time, and "to space birth" is about waiting for a certain period of time between pregnancies. The objective is to seek the understanding from respondent’s perspective on why the use of contraceptives 

Comment: What do you mean by "multiple responses"? indicate as a footnote under the table.

Response: Thank you for the enquiry. Multiple responses in the data collection refer to the situation where respondents are allowed to choose more than one answer or option for a particular question or item in a research questionnaire since multiple responses will better answer such questions. However, we have indicated it as a footnote under the table.

Comment: Table 3: Correct "Your source FP supply" to "Source of FP" What does "Not applicable" mean? "Just once" response under "how often do you use FP" does it mean the respondents have used FP only once in their lifetime? Please clarify. 

Response: Thank you for providing these insights, the comment “Your source of FP supply” in Table 3 has been corrected as you proposed. Also, “not applicable” refers to those participants who said that the FP methods ever used are calendar and Withdrawal methods. Since these methods are not sold in drug stores but rather based on personal decisions, the source of supply does not apply. Also, the response under "How often do you use FP" means the respondent has used FP only once at the time the data was collected.

Comment: Pages 13-14, lines 231-234: "Socio-cultural and health-related predictors of contraceptive use were minimal side effects (54.18%), counselling received on contraceptives (42.55%), partners support (36.73%), religious belief (36.36%), lack of knowledge (21.45%), the attitude of a service provider (26.18%), parental support (11.64%) etc (Fig 2)." This is not clear, please recast. What exactly is Figure 2 about, please explain. 

Response: Thank you for asking. On pages 13-14, lines 231-234, These are the socio-cultural and health-related factors that influence contraceptive use that was mentioned by the respondents apart from the demographic factors. These were multiple response types of data where respondents were asked to choose the various socio-cultural and health-related factors that can influence their contraceptive use. The statement has been recast as you proposed. Very grateful.

Comment: Page 14, lines 239-247: "As presented in Table 4, In a univariate analysis, the following independent variables were significantly associated;......lack of knowledge and counselling received." Please quote p values for every variable reported as significantly associated with contraceptive use.

Response: Thank you for the observation. On page 14, lines 249-253. The corresponding p-values have been reported.

Comment: Page 14, lines 247-248: "It was evident that Age was a factor in contraceptive use. older youth were 2.93 times more likely to use contraceptives than those at the adolescent stage." Please clarify which age groups were classified as "older youth" and "adolescent" 

Response: Thank you for notifying us, please "older youth" and "adolescent" is explained as “Older youth” (20-24) and "adolescent” (10-19) on page 14, lines 259 and 260.

Comment: Page 14, lines 250-253: "Respondents who were single were 92 times less likely to use contraceptives as compared to those who were married (AOR=0.08; 95%CI; 0.01-0.91), p=0.041 (Table 4). Respondents affiliated with the Islamic religion were 83 times less likely to use contraceptives as compared to those affiliated with Christianity (AOR=0.17; 95%CI; 0.05-0.64), p=0.003" Please reconcile the figures 92 and 83 quoted with the AOR.

Response: Thank you for providing these insights and we would like to appreciate you for an in-depth examination of our manuscript. We have rectified the comments on lines 257 and 260.

Comment: Table 4: How was "pressure to have sex" assessed?

Response: Thank you for the inquiry, Participants were asked about their encounters with situations where they felt compelled to participate in sexual activities, either by their romantic partner, acquaintances, or even family members, to satisfy certain desires or receive material benefits. So, the question was put “Do you feel pressured to have sexual intercourse”?

Discussion:

comment: Page 16, lines 272-274: "However, similar studies reported low contraceptive use among sexually active unmarried adolescent girls (15–19 years; 35.6%) and young women (20–24 years; 49.0%) [29]" Similar studies is mentioned by only one is cited/referenced. The adolescent age group is 10-19 and the youth "15-24"; please reconcile with 15-19 and 20-24 years quoted here.

Response: We are grateful for the observation. The statement “similar studies” have been rectified to “similar study” on line 289 and the age group has been reconciled to 15-19 and 20-24 on line 260, page 17.

Comment: The Discussion should be rewritten. Results should not be re-reported verbatim in the Discussion but discussed. Beyond simply comparing the results of this study with those of other studies, authors should explain possible reasons for their findings with relevant literature references. Results not initially reported in the results section should not be introduced for the first time in the Discussion

Response: Thank you for the comments on the discussion part of our manuscript. We are pleased to inform you that almost the entire discussion session has been rewritten.

Review Comments to the Author

Reviewer #1: 1. The title of the study needs to be modified to reflect the actual study population. The study was titled ‘Determinants and prevalence of modern contraceptive use among sexually active women’. This gives an impression that all sexually active women in reproductive age group should be part of the study. However, the Methodology and study in itself was limited to youths (people aged 15-24 years by United Nation’s definition).

Response: Thank you very much for your meticulous observation of our work, the title has been revised to “Determinants and prevalence of modern contraceptive use among sexually active youth in the Berekum East Municipality, Ghana”.

2. How does the findings from this study speak to a GLOBAL AUDIENCE with respect to contraceptive prevalence among young people?

Response: Thank you for the enquiry. The findings of this study on contraceptive prevalence among young people are of great significance to a global audience. With over 1.2 billion adolescents worldwide, the importance of effective contraceptive use cannot be overemphasized. The study reveals that there is a low uptake of modern contraceptive methods among young people, which puts them at risk of unintended pregnancies and unsafe abortions.

This issue is not confined to a specific region or country; it affects young people across the globe. By highlighting this problem, the study provides a global call to action for policymakers, healthcare providers, and educators to prioritize comprehensive sex education and access to modern contraceptive methods for young people.

The implications of this study go beyond just preventing unintended pregnancies. It speaks to the fundamental right of young people to make informed decisions about their sexual and reproductive health, and the need for them to have access to quality healthcare services that cater to their specific needs.

Therefore, the findings of this study should serve as a wake-up call to all stakeholders, regardless of geographic location, to prioritize and invest in programs and policies that promote the use of modern contraceptives among young people. Doing so will help to reduce the global burden of unintended pregnancies, unsafe abortions, and maternal mortality, and improve the overall health and well-being of young people.

3. The verbal interpretation of the odds ratio for Married respondents and Islamic faithful does not align with the data on the regression table. It was reported that Married youths and Islamic faithful were 92 and 83 times less likely to use contraceptives (respectively). This statement can’t be deduced from the table. Kindly reconcile

Response: We would like to appreciate you for an in-depth examination of our manuscript. We have rectified the comments on lines 261 and 264. Also, on page 16, line 277. The Table 3 heading has been edited as “Table 4: Univariate and multivariate regression model on determinants of modern contraceptive use among sexually active female youth”

4. The discussion was quite lacking in intellectual content. It appeared to be more of a repetition of the results. It will be good to revisit the discussion.

Response: Thank you for the comments on the discussion part of our manuscript. We are pleased to inform you that almost the entire discussion session has been rewritten.

5. The references were not properly written. They should be rewritten according to Vancouver guidelines. 

Response: Thank you once again for your time invested in our manuscript. The references have now been properly written in the Vancouver style.

CONCLUDING REMARKS: Again, thank you for allowing us to strengthen our manuscript with your valuable comments and queries. We have worked hard to incorporate your feedback and hope that these revisions persuade you to accept our submission.

---

## [Editor Report · Decision Letter 1]

19 May 2023

Determinants and prevalence of modern contraceptive use among sexually active female youth in the Berekum East Municipality, Ghana

PONE-D-23-04760R1

Dear Dr. Amoah,

We’re pleased to inform you that your manuscript has been judged scientifically suitable for publication and will be formally accepted for publication once it meets all outstanding technical requirements.

Kind regards,

Akaninyene Eseme Bernard Ubom, MBBS, MWACS, OMI Fellow

Academic Editor

PLOS ONE
---

## [Editor Report · Acceptance letter]

25 May 2023

PONE-D-23-04760R1 

Determinants and prevalence of modern contraceptive use among sexually active female youth in the Berekum East Municipality, Ghana 

Dear Dr. Amoah:

I'm pleased to inform you that your manuscript has been deemed suitable for publication in PLOS ONE. Congratulations! Your manuscript is now with our production department. 

Kind regards, 

on behalf of

Dr. Akaninyene Eseme Bernard Ubom 

Academic Editor

PLOS ONE